# Downregulation of mitochondrial metabolism is a driver for fast skeletal muscle loss during mouse aging

Raquel Fernando [1], Anastasia V. Shindyapina[2], Mario Ost [3,4], Didac Santesmasses [2], Yan Hu[2],
Alexander Tyshkovskiy[2,5], Sun Hee Yim [2,6], Jürgen Weiss[7,8], Vadim N. Gladyshev [2],
Tilman Grune [1,8,9,10,11,14 ✉] & José Pedro Castro [1,2,12,13,14]

Skeletal muscle aging is characterized by the loss of muscle mass, strength and function, mainly attributed to the atrophy of glycolytic fibers. Underlying mechanisms driving the skeletal muscle functional impairment are yet to be elucidated. To unbiasedly uncover its molecular mechanisms, we recurred to gene expression and metabolite profiling in a glycolytic muscle, *Extensor digitorum longus* (EDL), from young and aged C57BL/6JRj mice. Employing multi-omics approaches we found that the main age-related changes are connected to mitochondria, exhibiting a downregulation in mitochondrial processes. Consistent is the altered mitochondrial morphology. We further compared our mouse EDL aging signature with human data from the GTEx database, reinforcing the idea that our model may recapitulate muscle loss in humans. We are able to show that age-related mitochondrial downregulation is likely to be detrimental, as gene expression signatures from commonly used lifespan extending interventions displayed the opposite direction compared to our EDL aging signature.

[1] Department of Molecular Toxicology, German Institute of Human Nutrition Potsdam-Rehbrücke, 14558 Nuthetal, Germany. [2] Division of Genetics, Department of Medicine, Brigham and Women's Hospital, Harvard Medical School, Boston, MA 02115, USA. [3] Department of Physiology of Energy Metabolism, German Institute of Human Nutrition Potsdam-Rehbrücke, 14558 Nuthetal, Germany. [4] Paul-Flechsig-Institute of Neuropathology, University Clinic Leipzig, 04103 Leipzig, Germany. [5] Belozersky Institute of Physico-Chemical Biology, Moscow State University, Moscow 119234, Russia. [6] Department of Environmental Toxicology, Texas Tech University, Lubbock, TX 79401, USA. [7] German Center for Diabetes Research (DZD), Ingolstaedter Land Str. 1, 85764 Neuherberg, Germany. [8] German Diabetes Center (DDZ), Leibniz Center for Diabetes Research, Düsseldorf, Germany. [9] German Center for Cardiovascular Research (DZHK), 10117 Berlin, Germany. [10] University of Potsdam, Institute of Nutritional Science, 14558 Nuthetal, Germany. [11] Department of Physiological Chemistry, Faculty of Chemistry, University of Vienna, Vienna, Austria. [12] i3S, Instituto de Investigação e Inovação em Saúde, Universidade do Porto, 4200-135 Porto, Portugal. [13] Aging and Aneuploidy Laboratory, IBMC, Instituto de Biologia Molecular e Celular, Universidade do Porto, 4200-135 Porto, Portugal. [14] These authors contributed equally: Tilman Grune, José Pedro Castro. ✉email: scientific.director@dife.de

Aging is characterized by the accumulation of molecular defects throughout life, affecting all tissues. Skeletal muscle is one of the earliest tissues affected by age-related functional decline[1], which is characterized by the loss of muscle mass, strength and function, collectively known as sarcopenia. This phenomenon affects around 10% of the worldwide population (older than 60 years)[2]. This condition is associated with impaired locomotion, consequently, leading to a gradual lack of independence, which is correlated with morbidity, frailty and even mortality[3]. Depending on the type of muscle fiber, slow (type I) or fast (type II, which subdivides in type IIA, IIX, and IIB), different metabolic pathway strategies are employed. The former is enriched in mitochondria, displaying an oxidative metabolism, while fast fibers rely mainly in glycolytic metabolism[4]. Fast glycolytic fibers appear to be more prone to age-related alterations, and thus, their abundance declines over time[5–7]. However, the underlying mechanisms driving fast fiber shortage remain elusive. Type IIB fast fibers have been described as major regulators of whole-body metabolism, as they ameliorate the metabolism in obese mice, altering fatty acid oxidation in remote tissues[8] as well as reducing fat mass and hepatic steatosis in older animals[9], thus showing importance on combating these particular aspects of aging. Grounded on fast glycolytic muscle regulatory function on whole-body metabolism and its higher susceptibility to age-related changes, we focused on differences between young and old fast-twitch *Extensor digitorum longus* (EDL) muscle, where we did an extensive characterization of the main downregulated pathways from both RNA-seq and metabolomic analyses, which were related with mitochondrial processes, not only in aged mice but as well in elderly humans.

## Results

**Fast-twitch EDL muscle displays aging features but no fiber-type switch**. To unravel molecular differences between young and old (EDL) muscles, we performed RNA-sequencing as well as metabolomics of seven to eight muscle samples from 16- and 105-week-old C57BL/6JRj mice. From the RNA-seq analysis, 229 genes were found statistically differentially expressed in old compared to young EDL (Supplementary Table S1). Moreover, metabolite profiling revealed the same pattern of change, with 120 metabolites significantly changed in aged EDL muscle (Supplementary Table S2). In parallel with multi-omic analyses, we characterized a few parameters of skeletal muscle between young and old EDL. Interestingly, we did not observe loss of EDL muscle mass in old mice compared to young mice (Supplementary Fig. S1a). Adding to this, we also did not observe the canonical age-related switch in muscle fiber type, which maintained the same expression levels of myosin-heavy chains in young and old EDL, where *Myh4* (type IIB fiber marker) was the most abundant form (85% and 83%, respectively), further validated by qPCR (Fig. 1A). We further analyzed muscle fiber structure by using hematoxylin and eosin (H&E) staining. We observed that aged EDL fibers displayed apparent increased damage, accounting for segmental degenerated morphology or higher vacuolation (white spots inside the fibers) (Fig. 1B), as also observed in a previous study[10]. Although the phenotype appeared consistent, more images would be required for extensive quantification. Furthermore, we quantified muscle cross-sectional area (CSA), often used to characterize muscle atrophy at the histological level. In general, fibers from old mice displayed lower CSA compared to younger fibers, yet it did not reach statistical significance, except for fibers ranging from 1200 to 3999 μm$^2$, where older muscle fibers had a significant reduction of CSA (Fig. 1C). Moreover, at the molecular level, we saw a strong increase in 3-methylhistidine metabolite in older muscle, as well as MAFbx

(Atrogin-1) protein, a marker of protein breakdown and muscle atrophy, respectively[11] (Fig. 1D). At the gene expression level, there were increased levels of *Fbxo32* (MAFbx) and *Trim63* (MuRF1) (another marker of muscle atrophy), yet they did not reach significant differences (Supplementary Fig. S1b). Altogether, these results imply that no sarcopenia-like manifestations are present yet, however at the molecular level, markers of muscle damage are already increased in older muscles, suggesting that the ages employed for the analyses are in a non-sarcopenic or possibly in a pre-sarcopenic state, which is an important time point to characterize muscle molecular changes in order to understand the main trigger of the mechanisms of atrophy and further avoid loss of muscle mass.

**Non-sarcopenic mice display major downregulation of mitochondrial pathways in EDL muscle**. To acquire a comprehensive picture of the transcriptome signatures between young and old muscle, we performed gene expression profiling by RNA-seq of the two different age groups. We then applied gene set enrichment analysis (GSEA) to reveal age-related biological pathways over- and under-represented in old EDL muscles (Fig. 2A). The results showed that the upregulated pathways (activated) were related with synaptic signaling, neuron morphogenesis and an activation of immune response. The significantly downregulated pathways (suppressed) were found to be associated with mitochondria complexes and metabolism, such as mitochondrial complex chain assembly, mitochondrial gene expression, translation, electron transport chain and oxidative phosphorylation (Fig. 2A). As an example, we show a scheme of the electron transport chain, where all genes that code for the proteins of mitochondrial complexes are identified, including the ones that are downregulated in old EDL (highlighted in green) (Fig. 2B). To validate our findings and ascertain that up- and downregulated pathways were not due to another type of cells present in the skeletal muscle, as age-associated infiltrating cells, such as macrophages, fibroblasts or even adipocytes, we took advantage of public RNA-seq available data at the single cell resolution from mouse limb skeletal muscle. From all the present cells in the limb muscle (TSNE plot), we selected only skeletal muscle cells. Next, young and old cells were selected based on high expression of *Myh4* (*Myh4*$^{high}$) (Supplementary Fig. S2a). We then performed differential expression between young and old Myh4$^{high}$ cells followed by GSEA (Supplementary Fig. S2b). The results were similar to what we have obtained in aged EDL muscle signatures, namely activated pathways (upregulated), which were related with translational initiation, ribosome biogenesis, regulation of postsynaptic membrane potential, as well as with immune response. Similar to bulk aging EDL, suppressed pathways (downregulated) exhibited downregulation of oxidative phosphorylation, respiratory electron transport chain, among many others (Supplementary Fig. S2b). This emphasizes that the aging EDL signatures are mainly achieved from aged skeletal type IIB muscle cells. In addition to RNA-seq analysis, we performed metabolomic analysis to understand whether EDL metabolism would be affected during aging. To understand which metabolites were altered during aging, we inserted the significantly changed metabolites in Metabolist 5.0 (https://www.metaboanalyst.ca) which retrieved mitochondrial-related processes. Remarkably, the most downregulated metabolites compared to the young control group were found to be changed for mitochondrial components, such as mitochondrial respiratory chain complex I and NADH dehydrogenase complex, showing that aged EDL metabolome and transcriptome changes were related (Fig. 2C). Consistently, metabolites related to pathways responsible to generate ATP were found significantly lower, compared to younger ones. For

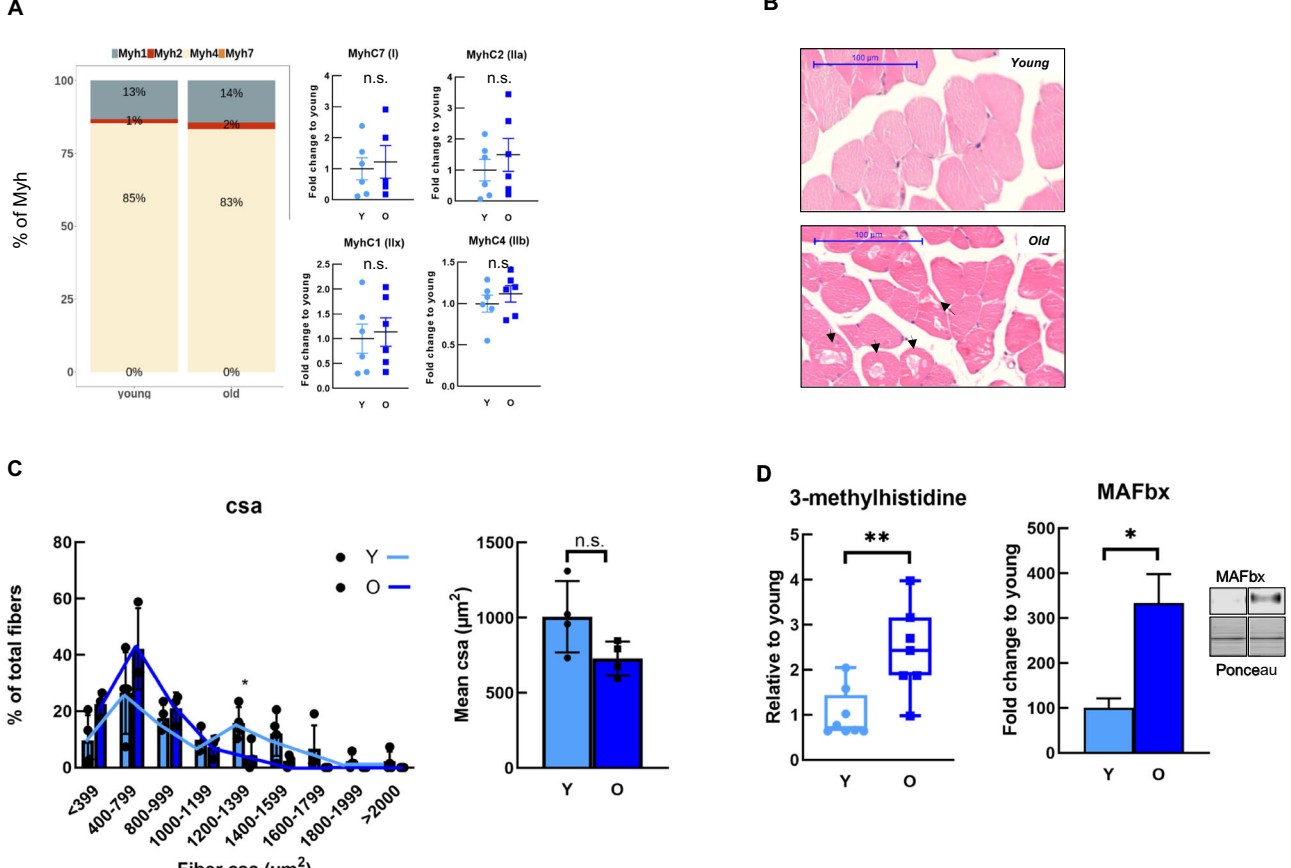

**Fig. 1 Fast twitching EDL muscle displays age-related changes with unaffected switch. A** The graph represents the percentage of Myosin-heavy chain (*Myh*) in young and old EDL muscle (on the left), where beige, red, gray and orange represent *Myh4*, *Myh2*, *Myh1*, and *Myh7* (the latter not graphically represented), respectively. The graphics on the right panel represent the different *Myh* gene expression levels, validated by quantitative PCR (*n* = 6–8 per group). **B** The H&E staining shows the cross-section EDL from young (upper image) and old (bottom image) mice, the latest with damaged fibers (black arrows). **C** The histogram shows the percentage of total fibers with different cross-sectional areas (CSA) (on the left) and on the right, the graphic represents the mean of CSA between young and old mice. **D** The left graphic depicts the levels of the 3-methylhistidine metabolite and on the right represents MAFbx protein levels, both show the fold change relative to young (*n* = 6–8). Statistical significance is given as follows *P < 0.05, **P < 0.01 (Mann–Whitney *U* test). Values presented as mean ± SEM; n.s. not significant.

example, NADH/NAD$^+$ ratio (0.74-fold and *P* = 0.0348), important cofactors in ATP production via anaerobic fermentation and aerobic respiration, was significantly decreased on older muscle (Fig. 2D). Intermediates of glycolysis as fructose-1,6-diphosphate were significantly declined, however 3-phosphoglycerate, 2-phosphoglycerate and phosphoenolpyruvate were significantly increased (Supplementary Table S3). Also, lactate level was significantly declined in old muscle (0.75-fold and *P* = 0.0052). Glucose and pyruvate levels decreased moderately in old animals (Fig. 2E). These are major products for ATP generation via glycolysis, however, our metabolite profiling platform does not measure the complete set of metabolites associated in this pathway. Adding to this, another important and fast via of ATP production is creatine phosphate metabolism, which was also affected, considering the significantly lower levels of creatine phosphate (0.34-fold and *P* = 0.0199) and creatine (0.96-fold and *P* = 0.0376) (Fig. 2F). The mitochondrial creatine kinase gene (*Ckmt2*), which codes for the enzyme that converts creatine into creatine phosphate was also significantly declined in old EDL at the gene expression level (Log2FC = −0.71 and *P* = 6.31 × 10$^{-14}$) (Supplementary Table S1). Together, these results indicate an energy level deficit in the aged EDL as many of the main energy pathways are impaired.

**Mitochondrial ultrastructure and respiration are affected during EDL aging.** To shed light on whether the revealed muscle energy deficit would already reflect changes in muscle morphology, we analyzed its ultrastructure by using transmission electron microscopy. In fact, several differences could be seen between young and old muscle ultrastructure (Fig. 3A). One striking visual effect that kept our attention was the net tangles, the so-called tubular aggregates (TAs) (Fig. 3A1), quantified in Fig. 3B. These aggregates consist of abnormal sarcoplasmic reticulum (SR) membranes, usually associated with mitochondrial activity impairment and accumulation of oxidative stress by-products in the myofibers[12] or with mutations in the stromal-interacting molecule-1 (*STIM1*) and Ca$^{2+}$ permeable channel of external membranes, *ORAI1* found in several muscle disorders such as TA myopathy (TAM)[13]. TAs are mainly associated with type IIB fibers[13], typically connected to muscle damage[12]. Consistent with this, was the increased oxidative stress marker, N6-Carboxymethyllysine, a known advanced glycation end product and the top upregulated metabolite in aged EDL (Fig. 3C). Although, *Stim1* (located inside the SR) and *Orai1* (located in transverse tubules) were not changed during aging in our RNA-seq dataset, *Casq1*, a Ca$^{2+}$ binding protein (located inside the cisternae lumen) was found to be significantly downregulated (Fig. 3D).

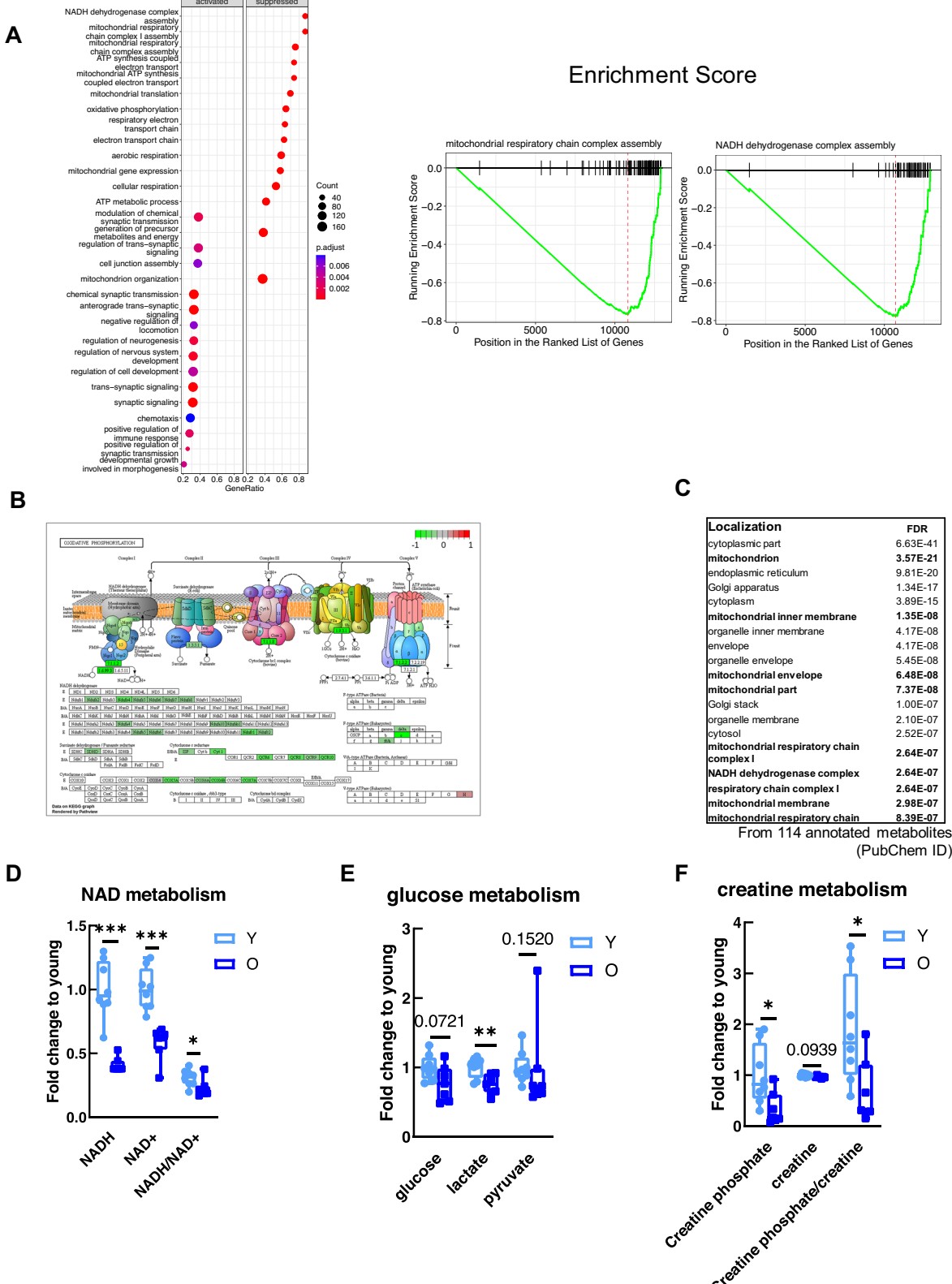

The extracellular matrix was also thicker, possibly due to accumulation of collagen during aging (Fig. 3A2, old). In fact, an increased amount of collagen was also observed in the trichrome staining, yet it did not reach statistical significance (Fig. 3E). Strikingly, older fibers also displayed several vacuoles (Supplementary Fig. S3a1), as well as swollen sarcoplasmic reticulum surrounding mitochondria (Supplementary Fig. S3a2). Those

observations are in line with the significantly lower levels in spermidine metabolite (0.31-fold and $P = 0.0005$), (Supplementary Table S3), known to induce autophagy (Fig. 3F), as well as the transcript levels of the enzyme that converts spermine into spermidine, the spermidine oxidase *Smox* (Log2FC $= -1.55$ and $P = 1.6 \times 10^{-5}$) (Fig. 3F and Supplementary Table S1), suggesting a decline in autophagy. Consequently, the results imply that

**Fig. 2 Non-sarcopenic mice display major downregulation of mitochondrial pathways in aged EDL muscle. A** Gene Set Enrichment Analysis (GSEA). The left side shows the biological processes that are activated and on the right side the ones which are suppressed. The graphics on the right represent the enrichment score of GO term from mitochondrial respiratory chain complex assembly. **B** Schematic representation of electron transport chain with the associated genes. Downregulated genes in aged EDL related to mitochondrial complexes are highlighted in green. **C** Associated localization of significantly downregulated metabolites from 114 annotated metabolites. **D** NAD metabolism. The graphics show the fold change of NAD reduced, oxidized and the ratio between young and old EDL muscle. **E** Glucose metabolism. The graphic shows the levels of glucose as well as two products of glycolysis (lactate and pyruvate), between young and old muscle. **F** Creatine metabolism. All values are relative to the mean of young controls, except the fold change ratios. Statistical significance is given as follows *$P < 0.05$, **$P < 0.01$, ***$P < 0.001$ ($t$ test). Values presented as mean ± SEM.

mitophagy may be impaired as well, indicated by the significantly increased levels of Parkin protein with unchanged gene expression (Supplementary Fig. S3b), and it is known that this E3 ligase is induced by the accumulation of misfolded mitochondrial proteins[14]. Mitochondria can also be seen surrounding autophagosomes, indicating that they might be dysfunctional, being in the process to be degraded (Supplemental Fig. S3a1). In parallel with this was the apparent loss of mitochondrial integrity in older fibers, indicated by the loss of mitochondrial cristae in intermyofibrillar mitochondria, as shown in Fig. 3A3, old. Sub-sarcolemmal mitochondria were enlarged and clustered in the extracellular matrix area (Fig. 3A2, old), a feature not observed in younger muscle fibers (Fig. 3A3, young). Nevertheless, mitochondrial dynamics machinery, such as Drp1, Fis1, and Mfn2 did not change. To better understand the nature of the predicted mitochondrial impairment, we measured mitochondrial respiratory capacity per tissue wet weight in young and old EDL muscles. Here, saponin-permeabilized fibers were subjected to a multiple substrate-uncoupler-inhibition titration (SUIT) protocol, used to measure high-resolution respirometry[15] (Fig. 3G). In general, oxygen consumption rate (OCR) was diminished in aged EDL muscles in different respiration states, for instance, in the OXPHOS capacity and maximum electron transfer system (ETS). Also, the non-phosphorylating LEAK respiration (oxygen flux without ADP, but with addition of malate and pyruvate complex I substrates) was lower in the old EDL muscle. Further, the addition of ADP to saturate OXPHOS capacity and maximum coupled respiration from both complex I and II (CI and CII, after the addition of glutamate and succinate), was found to be significantly lower in aged EDL (Fig. 3G). Next, after the addition of an exogenous uncoupler, FCCP, to collapse the proton gradient, the maximum electron transfer system capacity (ETS CI&CII) was found to be diminished again in the aged EDL muscle. Finally, and as a consequence of complex I inhibition with rotenone, the submaximal ETS CII respiratory state, was also reduced in older EDL muscle. Taken together, these results suggest mitochondrial respiration dysfunction might be the ground cause of mitochondrial degenerated morphology, as mitochondrial respiration impairment, can increase ROS leakage, increasing chronic oxidative stress, leading to the formation of oxidative stress markers and consequently changes in cell organelles and morphology, for instance the TAs.

**Mouse EDL age-related signatures overlap with skeletal muscle human aging**. To complement our analysis, we further investigated whether the same pathways would be affected in human aging. Therefore, we took advantage of human mixed-fiber *Gastrocnemius* muscle data from the Genotype-Tissue Expression dataset (GTEx). We crossed old EDL data (mouse aging) with male and female human aging (across different ages) (Supplementary Fig. S4). Interestingly when comparing a human young group to an elderly group (22–29 years vs 70–79 years, respectively), we found 22 downregulated and 55 upregulated common pathways to aged EDL muscle (mouse aging) (Supplementary Fig. S4). Next, we performed GSEA to retrieve enriched pathways common to mouse

and human muscle aging and observed consistent results (Fig. 4A). This includes pathways associated with mitochondrial and energy metabolism, as electron transport chain, cellular respiration and mitochondrial gene expression. The pattern is similar for males and females and shows an overall intensification in downregulation (Fig. 4A). Upregulated pathways were again similar between sexes, mostly related to immune response, regulation of postsynaptic membrane potential and neuron projection development. The graphics below the heatmaps show that downregulated (purple) and upregulated (orange) genes are strongly correlated with aging (Fig. 4B), and the most correlated genes, SLC25A20 and SLC25A51 are shown for female (Fig. 4B, left) and male (Fig. 4B, right), respectively. Other important downregulated pathways that correlated with aging included pathways associated with protein transmembrane transport, mitochondrial gene expression, the establishment of protein localization to the mitochondrion, cytochrome complex assembly, and cofactor biosynthetic process (Supplementary Fig. S5). On the other hand, upregulated pathways during human aging were among others, related with axon development, cell morphogenesis involved in neuron differentiation, muscle system process, muscle cell differentiation, and synapse organization (Supplementary Fig. S5).

**Lifespan-extending interventions are negatively associated with aging EDL muscle**. Based on these results, we wanted to understand whether the observed mitochondrial and neuromuscular junction changes would be detrimental, neutral or adaptive. Thus, we compared the transcriptome profile from aged mouse EDL with those induced by known longevity interventions, such as growth hormone deficiency, caloric restriction (CR), rapamycin, among others[16], as well with aging signatures of mice, rats and humans, identified from multiple publicly available datasets[17]. We found that aged EDL gene signatures were negatively associated with biomarkers of lifespan-extending interventions, including CR and rapamycin, but in parallel, were positively associated with transcriptomic age-related changes occurring in multiple organs, such as brain, muscle and liver, as well as across tissues in mouse, rat and human (Fig. 5A). Utilizing GSEA, we examined functional pathways associated with the mammalian aging and the effect of longevity interventions in mice (Fig. 5B). The heatmap reflects mitochondrial-related processes and pathways, including mitochondrial respiratory chain complex assembly, oxidative phosphorylation and mitochondrial translation, that are strongly downregulated in aged EDL muscle and other organs during aging, but are upregulated in response to lifespan-extending interventions. At the same time, upregulated processes in aged EDL, such as immune response and interferon signaling (alpha and gamma response), were generally attenuated by most longevity interventions and were negatively associated with mouse lifespan. Together, these results show that different interventions, such as caloric restriction and rapamycin, can counteract the effects of aging in multiple tissues, including EDL muscles, and that muscle age-associated gene expression changes related to mitochondrial function and immune response seem rather detrimental than neutral or adaptive.

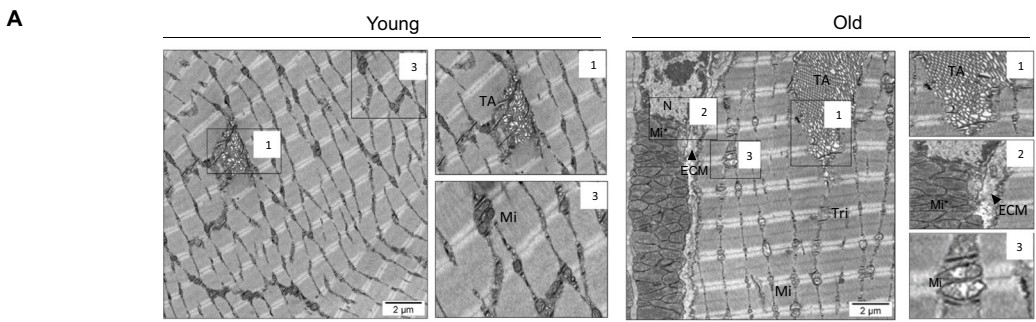

**Fi**-Muscle Fiber  **N** –Nucleus  **ECM**-Extracellular Matrix  **Mi**-Mitochondria (IMF)  **Mi\***-Mitochondria (SSL)
**TA**-Tubular Aggregates  **Tri**-Triade  **Sr**- Sarcoplasmatic reticulum swollen

## Discussion

Aging is detrimental for human body functionality. Particularly, the aging process induces a progressive decline in skeletal muscle function (including loss of mass and strength). It is well known by the literature that fast glycolytic muscle fibers are more susceptible to age-related changes. In the present study, we set out to understand why the fast glycolytic muscle is highly susceptible to age-related changes and what are the underlying metabolic mechanisms[18,19]. Here, we could demonstrate that old fast gly-colytic muscle fibers undergo several different morphological, transcriptomic, and metabolomic changes, associated with muscle damage markers, energy metabolism, neuromuscular junctions (NMJ) and immune functions. Importantly, mouse limb skeletal muscle available data on RNA-seq from selected *Myh4* positive

**Fig. 3 Mitochondrial structure and respiration capacity are affected during EDL aging. A** Representative electron microscopy images of young and old EDL muscle. Few characteristics are depicted, numbered and zoomed in ($n = 3$ per group). **B** Area of tubular aggregates quantified per TEM picture as fold change relative to young ($n = 3$ animals per group; 8–12 pictures were analyzed per group). Values presented as mean ± SEM. **C** N6-carboxymethyllysine metabolite levels. Fold change relative to young. **D** Sarcoplasmic (SR)-related genes with Log2Fold change relative to young. **E** Trichrome (TC) staining. Area quantification performed with image J. Fold change relative to young. **F** Log2Fold change of spermidine oxidase (*Smox*) and spermidine metabolite. Fold change relative to young. Statistical significance is given as follows \*$P < 0.05$, \*\*$P < 0.01$ (Mann–Whitney U test). **G** Representative analysis of mitochondrial respiration from young and old EDL muscle. Quantification of oxygen consumption rate (OCR) in pmol/min/µg of protein ($n = 5$ per group). Values presented as mean ± SEM, n.s. not significant.

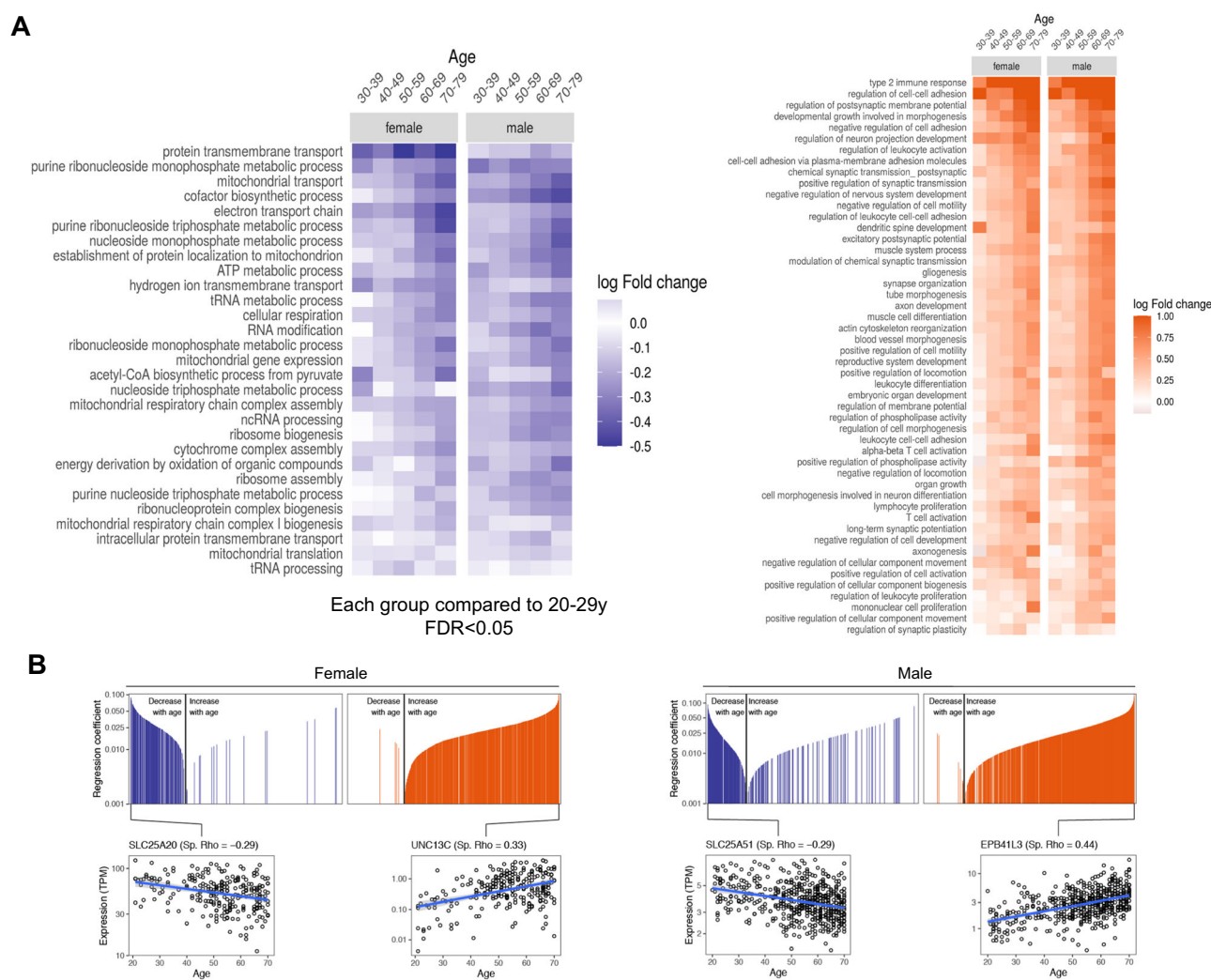

**Fig. 4 Mouse EDL age-related signatures overlap with skeletal muscle human aging. A** Functional enrichment across different ages in female (left panel) and male (right panel) humans. The heatmap represents the log-fold change and each group is relative to the youngest group with ages between 20 and 29 years old. On the left the heatmap shows the downregulated biological processes, in purple, and on the right, the upregulated biological processes during aging, in orange, FDR < 0.05. **B** The graphics show all the downregulated (in purple) or upregulated (in orange) biological processes that are decreasing with aging (on the left part of each graphic) or increasing with aging (right part of each graphic). Each single line in the upper graphics represents the correlation of the expression of one gene during aging. The four graphics below represent the genes most positively or negatively correlated with aging in female (on the left) and in males (on the right).

cells, revealed identical up and downregulated biological processes, guaranteeing the specificity of our EDL muscle data.

In contrast to what has been previously described[10], we did not observe fiber switching in aged muscle. This difference may be related to the muscle fiber-type composition used in each study, as mixed-fiber-type muscles have more content in hybrid fibers, which during aging function towards a more oxidative metabolism compared to muscles highly enriched on specific fiber types, as EDL muscle. Nevertheless, older fibers displayed lower CSA

compared to younger ones. This has also been previously observed in other hindlimb muscles from aged mice[10,20] and humans[21]. A possible explanation for CSA decline in aged EDL could be the increased muscle atrophy, considering the increased levels of muscle degradation (3-methylhistidine) and atrophy (MAFbx) markers. Another feature supporting muscle damage was the upregulation of immune response, revealed by gene set enrichment analyses. This is not surprising, as aged tissues are usually characterized by an increased infiltration of immune cells,

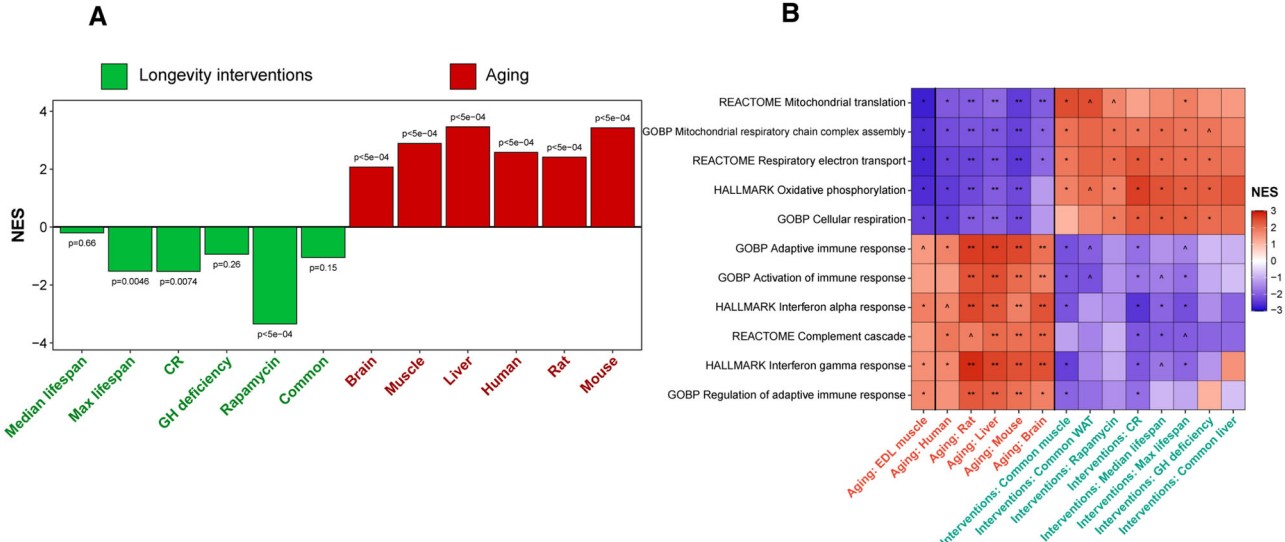

**Fig. 5 Lifespan-extending interventions are negatively associated with aging EDL muscle. A** The graphic represents the negative and positive associations between aged EDL gene signatures and signatures of longevity interventions (left) and aging (right), respectively. *P* value for association with each signature was assessed using GSEA-based test. **B** Functional enrichment analyses (GSEA) of gene expression profile of aged EDL muscle (left), signatures of mammalian aging (middle) and signatures of lifespan-extending interventions in mice (right). Cells are colored based on the normalized enrichment score (NES). The whole list of enriched functions is in Supplementary Table S3. ^*P*.adjusted<0.1; *P.adjusted <0.05; **P.adjusted <0.01; ***P.adjusted <0.001.

such as macrophages, which, consequently, are responsible to induce pro-inflammatory cytokines[22,23]. Tumor necrosis factor (TNF-a) and interleukin 6 (IL-6) are known to induce muscle atrophy[23]. TNFα can activate NF-KB signaling pathway, leading to MuRF1 overexpression and muscle atrophy[24], however, transcript levels of these cytokines were not increased in our study. Another important marker contributing to muscle atrophy is forkhead box O3 (Foxo3), by enhancing *Trim63* expression, which was upregulated in aged muscle, close to reach significance (Log2FC = 1.06 and *P* = 0.078) (Supplementary Table S1). This may justify the increased levels of E3 ligases[25]. Furthermore, GSEA showed upregulation of other biological processes, such as synaptic signaling and neuron projection morphogenesis in older muscle. These results drive us to speculate about a possible compensation of the NMJ system, accounting for the higher transcript levels of some acetylcholine receptors (AchR), such as *Chrna1* (Log2FC 0.39; *P* = 0.029) and *Chrnb1* (Log2FC 0.45; *P* = 0.067) (both Supplementary Table S1), to maintain the stimuli for muscle contraction, since at this stage the cells appear to be functional impaired. In addition, the levels of the neurotransmitter histamine were elevated in aged EDL (1.66-fold, Supplementary Table S3). Studies have shown that the skeletal muscle goes through transient "denervation-reinnervation" cycles during aging that can modify muscle fiber types[26]. However, during aging, the mechanisms of reinnervation begin to fail and denervation becomes prevalent, contributing to the acceleration of muscle atrophy and thus enrichment of slow muscle fibers[26,27]. Nevertheless, downregulated pathways were mainly related with mitochondrial processes which may suggest the relevant role of intracellular mitochondrial dysfunction prior to established phenotypes of denervation and muscle atrophy. Yet, during aging, mobility and locomotion become progressively affected due to increased sedentarism. In fact, mitochondrial respiration was affected in aged EDL, as also observed in *Vastus lateralis* from pre-frail elderly subjects and only rescued in physically active subjects[28]. The decline in mitochondrial respiratory capacity could either be due to the lower expression of protein complexes formation, or lower levels of cofactors necessary to run

mitochondrial respiration. Notably, while there were no changes in mitochondrial complexes, metabolomic analyses showed a decline in the NADH/NAD⁺ ratio in older EDL muscle. The lack of these cofactors appears to be key in the decline of mitochondrial energy production, as they are widely involved in driving ATP synthesis. For instance, the electron transport chain allows electron flux, from the oxidation of cofactors such as NADH and FADH. Remarkably, NAD⁺ was strongly downregulated in human-aged muscle, however, the levels could be recovered by exercise in elderly subjects[29]. Interestingly, NAD⁺ was also positively associated with mitochondrial respiration, demonstrating the association between NAD⁺ and healthy aging[29]. Features such as the loss of mitochondria cristae, enlargement and agglomeration of SSM near the extracellular matrix were observed in the current study and also reported by Leduc-Gaudet et al.[30] This phenomenon could be used as a mechanism to conserve energy, since older muscle displays a more rigid structure, avoiding mitochondria to be dynamic as in younger muscles. However, no changes were found in mitochondrial fission and fusion machinery, as usually observed in older muscle[31].

Another remarkable observation was the increased area of tubular aggregates in aged muscle, known to be formed by irregular proliferation of sarcoplasmic reticulum (SR) and thus destabilizing the connection links involving SR with tubular system and myofibrils, which may help to the formation of a more rigid structure in older tissue. Interestingly, tubular aggregates can be less prominent when aged mice are supplemented with resveratrol[32] as well as in trained old mice compared with non-trained[13]. In parallel with the observation of TAs, were the presence of vacuoles, surrounded by damaged mitochondria. Parkin (E3 ligase), involved in the ubiquitination of proteins and recruited to the outer mitochondrial membrane, was highly expressed at protein level in older EDL, with no changes in mRNA levels, suggesting an impaired degradation with a consequent protein accumulation. Furthermore, an important inducer of autophagy, spermidine[33], was also significantly decreased in our metabolic analyses, possibly attributed to the decline in spermidine oxidase expression (Log2FC = −1.55 and *P* = 1.6E⁻⁵)

(Supplementary Table S1), which is in line with the findings from ref. [34]. Together, these results suggest an impaired autophagy, which we have previously documented in old EDL, shown by the higher levels of LC3-II, accumulation of p62, and declined lysosomal activity[35].

To further investigate whether the same pathways in aged EDL muscle were similar in humans, we combined mouse EDL RNA-seq data with human *Gastrocnemius* RNA-seq obtained from the GTEx database. Consistently with mouse-aged EDL data, human *Gastrocnemius* muscle also revealed common affected pathways. This is of the utmost importance, showing that similarly to mouse muscle aging, human muscle aging also lacks energy, possibly due to mitochondrial impairment from fast glycolytic muscle.

Grounded on mitochondrial morphology and function alterations in mouse and human aging, we sought to understand whether these changes would be neutral, adaptive or detrimental. Therefore, we resorted to disclose the association of aging EDL signatures with features of multiple lifespan-extending interventions, such as caloric restriction, growth hormone deficiency or rapamycin, in liver, muscle and white adipose tissue. Remarkably, the results suggest excessive immune response and mitochondrial energy processes alterations to be detrimental during aging since ATP production by mitochondria is crucial for cell function and this can be ameliorated by longevity interventions. Nonetheless, synaptic signaling may be an adaptation in aging tissue, to cope with a lower input of muscle stimuli during aging, and therefore the system compensates with increasing synapsis, as there is loss of motor neurons, although with motor unit remodeling[26,36]. These findings are in accordance with Uchitomi and colleagues, since they found an increased amount of neurotransmitter levels (acetylcholine, histamine and serotonin) in aged glycolytic fibers[20].

Overall, compared to previous studies using either skeletal muscle transcriptomics or metabolomics approaches individually, the present study brings the combination of both analyses from young and old mice glycolytic fibers. The analyses revealed several differences between young and old fast glycolytic fibers regarding neuromuscular junction, immune response and especially to energy pathways. Furthermore, the combination of computational and experimental analyses highlights a major dysregulation of mitochondrial function in aged glycolytic muscle. This study shares strong evidence that skeletal muscle aging is highly associated with mitochondrial impairment in fast glycolytic fibers, detrimental for muscle functionality. However, it remains to be elucidated whether mitochondrial dysfunction is the trigger for upregulation of synaptic signaling as a compensation of the neuromuscular junction system. Hence, it might be of interest to investigate this mechanism in detail in further studies. In addition, we provide new insights that can be used in further studies, focusing on fast glycolytic muscle fiber types, that can improve muscle energy status, ameliorating not only lifespan but healthspan as well.

## Methods

**Animals**. Male C57BL/6JRj mice were purchased from Janvier Labs (France), at 16 and 105 weeks of age. Mice were euthanized with isoflurane followed by cardiac puncture. Skeletal muscle was collected and stored accordingly until processed. All experimental procedures were performed in accordance with the guidelines of German Law on the Protection of Animals and were approved by the local authorities (Landesamt für Umwelt, Gesundheit und Verbraucherschutz, Brandenburg, Germany).

**Eosin and hematoxylin staining**. The paraffin-embedded EDL were cut into 2-μm cross-sectional slices. The paraffin-embedded slices were rehydrated by subsequent washes of Roti-Histol (Carl Roth; 6640) and ethanol. Further, the slices were stained with Hematoxylin-eosin (H&E) staining (Sigma-Aldrich; GHS316, Sigma-Aldrich; HT110232, respectively). The images were obtained using the MIRAX scanner (Zeiss) and quantification of the cross-sectional area was performed using the ImageJ program.

**RNA isolation, RNA-sequencing, and RNA analysis**. RNA isolation and quality control, library construction and RNA-seq analysis was done by OakLabs GmbH (Hennigsdorf, Germany). Sequencing was performed with NextSeq 500 with 75 bp single-end reads. 30 M read depth coverage per sample were used. Fastq files were mapped to the mouse genome and gene counts were achieved with START v.2.7.2b[37]. Statistical analyses were performed in R[38]. GSEA was performed using the cluster profile package in R[39] and ranks for each gene were calculated as $-\log10$ (P value) multiplied either by -1 if log-fold change was negative or by 1 if positive.

**Association with gene expression signatures of aging and lifespan-extending interventions**. Association of gene expression log-fold changes induced with age in mouse EDL muscles with previously established transcriptomic signatures of aging[17] and lifespan-extending interventions[16] was examined as described in ref. [17]. Signatures of lifespan-extending interventions included genes differentially expressed in mouse tissues in response to individual interventions (Fig. 5A), along with common patterns of lifespan-extending interventions, and expression changes associated with the intervention effect on mouse maximum (Max lifespan) and median lifespan (Median lifespan).

For the identification of enriched functions for signature of aged EDL muscle, signatures of aging and signatures of lifespan-extending interventions, we performed functional GSEA[40] on a pre-ranked list of genes based on $\log_{10}$(P value) corrected by the sign of regulation. GO BP, REACTOME, KEGG and HALLMARK ontologies from the Molecular Signature Database (MSigDB) were used as gene sets for GSEA. The GSEA algorithm was performed separately for each signature via the *fgsea* package in R with 5000 permutations. To adjust for multiple testing, we performed a Benjamini-Hochberg correction[41]. An adjusted P value cutoff of 0.1 was used to select statistically significant functions. A heatmap colored by NES was built for manually chosen statistically significant functions (adjusted P value < 0.1) (Fig. 5B). Complete list of functions enriched by at least one signature is included in Supplementary Table S4.

**Metabolomics**. Metabolome analysis was performed by Metabolon® (North Carolina, USA). The raw data was extracted, peak-identified and quality control processed using Metabolon's hardware and software. Compounds were identified by comparison to library entries of purified standards or recurrent unknown entities.

The targeted metabolite profiling platform measured 505 metabolites, including lipids. First, the total ion current (TIC) values of each metabolite were examined in order to eliminate false values. Further, metabolites that are significantly different in any of the two groups were selected using a wide array of commonly used statistical and machine learning methods. The initial filtering of the TIC values of each metabolite resulted in the identification of 120 metabolites different between ages (P < 0.05).

The metabolites were analyzed further using MetaboAnalyst 5.0[42] in order to identify molecular pathways, metabolite sets enrichment, and network. The data were further visualized to provide a context for gene regulation.

**mRNA isolation and real time-qPCR.** mRNA extraction from EDL muscle lysates was performed using Dynabeads mRNA DIRECT Kit (ThermoFisher Scientific no.61012) and transcribed to cDNA with SensiFAST cDNA Synthesis Kit (Bioline, no.BIO-65054), to the instructions of the manufacturer. Quantitative real time-PCR (qPCR) was performed in the presence of 1X DreamTaq Buffer, DreamTaq Hot Start DNA Polymerase (ThermoFisher Scientific, no. 15619374), 2 mM dNTPs, 1x SYBR Green (Qiagen no.330502) and 1 µM of forward and reverse primers. *Tbp*, *Rpl13a*, and *Actin* were used as internal normalization controls. Primer sequences are referred to in Supplementary Table S5.

**Immunoblotting.** EDL muscles were homogenized with a tissue lyser in RIPA lysis buffer, supplemented with phosphatase and protease inhibitors (PhosStop, no. 4906845001, Sigma-Aldrich and Protease inhibitor Cocktail, no. P8340, Sigma-Aldrich, respectively). Further, the lysates were centrifuged at 14,000 rpm at 4 °C for 10 min and supernatants were collected. Protein concentration was determined by Lowry assay. In total, 10–15 µg of protein were separated in sodium-dodecyl sulfate-poly-acrylamide gel electrophoresis (SDS-PAGE) and transferred to nitrocellulose membranes. Afterward, membranes were incubated with primary antibodies overnight at 4 °C, followed by incubation with fluorescent-labeled secondary antibodies. Immunodetection was performed by Li-Cor Biosciences equipment from Odyssey. Primary and secondary antibodies are listed in Supplementary Table S6.

**Mitochondrial respiration.** High-resolution respirometry was measured in permeabilized EDL muscle fibers, using a high-resolution Oxygraph-2k (OROBOROS Instruments, Innsbruck, Austria), following the protocol previously described[15]. Briefly, muscles were dissected and placed in ice-cold biopsy preservation medium (BIOPS). Next, EDL fibers were separated and permeabilized with saponin and washed with mitochondrial respiration medium Respiratory capacity was analyzed by performing a multiple substrate-uncoupler-inhibitor titration (SUIT) protocol[15]. Data were normalized to muscle wet weight and Oxygen Consumption Rate (OCR) was calculated by pmol.O$_2$.mg fiber.

**Transmission electron microscopy.** Skeletal muscle tissues (EDL) were collected and fixed for 2 h at room temperature by immersion in 2.5% glutaraldehyde in 0.19 M sodium cacodylate buffer, pH 7.4. Further, they were postfixed in 1% reduced osmium tetroxide in aqua bidest for 90 min, and subsequently stained with 2% uranyl acetate in maleate buffer, pH 4.7. The specimens were dehydrated in graded ethanols and embedded in epoxy resin, according to Spurr, A[43]. Ultrathin sections were cut in a longitudinal direction. Sections were picked up onto Formvarcarbon-coated grids, stained with lead citrate, and viewed in a transmission electron microscope (TEM 910; Zeiss, Germany). Micrographs were taken randomly.

**Quantification and statistical analysis.** Statistical analysis and graphics related with qPCR, immunoblotting and mitochondrial respirometry were performed using GraphPad Prism 8 Software. Shapiro–Wilk test was used to determine the normal distribution of the tested variables of the mentioned methods. Statistical details for each experiment are provided in the figure legend. Statistically significant differences were considered when $P \leq 0.05$.

**Reporting summary.** Further information on research design is available in the Nature Portfolio Reporting Summary linked to this article.

## Data availability

Numerical source data for all graphs in the manuscript can be found in the supplementary data file. Data are available in the supplementary material (Supplementary_Material_File.xlxs and Supplementary_Data_File.pdf and Supplementary_Material_2.pdf). The accession code for RNA-seq deposited data is PRJNA1035515 on Sequence Read Archive (SRA) under the National Library of Medicine.

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

## Acknowledgements

The authors would like to thank to Stefanie Deubel and Tanina Flore for their technical support. The work was supported by the German Center for Diabetes Research (DZD) and the German Centre for Cardiovascular Research (DZHK).

## Author contributions

R.F., J.P.C., and T.G. formulated the hypothesis and designed the experiments. R.F. performed most of the experiments. M.O. performed OROBOROS experiment. J.W. performed TEM analyses. A.S., D.S., and A.T. performed the RNA-seq analyses. S.H.Y. performed metabolomics analyses. R.F. wrote the manuscript draft. J.P.C. reviewed the manuscript. J.P.C., M.O., V.G., and T.G. proofread the manuscript.

## Funding

## Competing interests

The authors declare no competing interests.
