## [Peer Review File · Communications Biology]

Reviewers' comments:

Reviewer #1 (Remarks to the Author):

Fernando et al., submitted the manuscript, "Downregulation of mitochondrial metabolism is a driver for fast skeletal muscle loss during mouse aging", gave us the information about that the features of aged EDL (mostly including type IIB myofibers). The authors presented different analyzed data to show the possible mechanisms and regulation in EDL during aging; however, most observations were already known and published in previous studies. The authors need to show the novelty in their study. There are several comments for this study.

1. As the title shown, the authors want to find the mechanism of fast myofiber loss during aging. Data shown EDL had mild tubular aggregation and may have smaller myofiber size, but no loss of muscle mass. Have any other evidence to support the EDL muscle loss during aging?
2. The old mice used in this study were 105 weeks (over 24 months old) should have severe sarcopenic phenotypes. Why authors mention that is non-sarcopenic mice?
3. This manuscript indicated that the dysfunction of mitochondrial respiration was due to the morphological defect of mitochondria. (1) The defects of mitochondrial morphology have bad effects on OXPHOS and glycolysis function, as well as most metabolic actions in mitochondria. (2) authors chose EDL to be the studied target. Most myofibers in EDL are type II. Why authors want to focus on the OXPHOS function? Have any special reason?
4. In this manuscript, authors compared the gene expression between mouse EDL and human gastrocnemius and found that they had similar expression patterns. However, these two parts of muscles were composed of different types of myofibers. If they had similar expression patterns at aged stage, dose it has any meaning? Similarly, why different aged tissues/organs were positively associated with transcriptomic age-related changes? Different tissues/organs have different functions and regulation. If they have similar age-related changes, dose it has any meaning?

Reviewer #2 (Remarks to the Author):

Sarcopenia is mainly attributable to the atrophy of fast glycolytic fibers. Fernando et al. used gene expression and metabolite profiling of the extensor digitorum longus (EDL) muscles from 16- and 105-week old mice. Changes based on transcriptomic and metabolomic profile were further validated using various assays oroboros mitochondria respiration, immunoblot, transmission electron microscopy). Commendably, additional public data sets were included in analyses. scRNAseq of Myh4 expressing nuclei was used to confirm that changes measured in intact EDL muscle are consistent with nuclei from type 2b fibers. Other publicly available datasets from human samples of various ages were compared with mouse datasets offering insights into the translatability of their findings with overlapping pathways. Interpretation was further extended by showing that life span extending interventions are negatively associated with changes occurring in EDL muscle. The results mostly confirm the findings of various individual studies across the literature but with the combination of metabolomic and transcriptomic analyses. The use of publicly available datasets to extend the findings was a strength of the study.

Minor comments:

- Changes in muscle mass would be better expressed normalized to TA length.
- Fast muscle fibers are not 'resistant to power and strength'
- MuRF1, Mafbx are better described as markers of atrophy than 'damage'. Damage frequently implies injury to cell membrane rather than breakdown of protein
- Discussion states that this study set out to understand why fast glycolytic muscle fibers is susceptible to aging relative to slow oxidative muscle but the experiments did not compare fast and slow muscle.

Referee expertise:

Referee #1: Aging + muscle + transcriptomics

Referee #2: Skeletal muscle health

Reviewers' comments:

Reviewer #1 (Remarks to the Author):

Fernando et al., submitted the manuscript, "Downregulation of mitochondrial metabolism is a driver for fast skeletal muscle loss during mouse aging", gave us the information about that the features of aged EDL (mostly including type IIB myofibers). The authors presented different analyzed data to show the possible mechanisms and regulation in EDL during aging; however, most observations were already known and published in previous studies.

The authors need to show the novelty in their study. There are several comments for this study.

We thank the reviewer for taking the time to go through our manuscript and provide valuable input that will certainly improve the final version. We acknowledge that the implication of mitochondria in muscle aging has been previously evaluated by others. However, our study generated a unique validation set-up of results. We have focused our study in the fast-twitch skeletal muscle EDL which has not been clearly associated with mitochondrial function decline with aging since this fiber type is mostly glycolytic. We established this rather unrecognized link in our study originated from unbiased approaches such as transcriptomics (Fig. 2A) and metabolomics (Fig. 2C) having had the downstream analysis the purpose to validate mitochondrial processes dysfunction, employing a whole range of assays ranging from mitochondrial ultrastructure (Fig. 3A and Fig.S3A) to mitochondrial respiration (Fig. 3G). Moreover, many processes change with age, but the question whether these reflect adaptations or are in fact detrimental are rarely exploring in aging research papers. To understand where these mitochondrial alterations in EDL and in terms of adaptation or deleterious effects, we crossed our EDL gene expression signature with aging signatures (multi-tissue, multi-species) and with interventions described to extend lifespan such as CR or rapamycin (Tyshkovskiy, A *et al.*, *Cell Metab.*, 2019; Flanagan E.W *et al.*, *Annu Rev Nutr.*, 2020; Shindyapina A.V *et al.*, *Sci. Adv.*, 2022). We found the positive association with aging signatures, suggesting some overlap of pathways in aging tissues and conservation across species, and negative association with longevity interventions (Fig. 5) which strongly suggests that the mitochondrial changes we observe with EDL aging are in fact detrimental. Also, we translated our finding to human muscle aging using GTEx dataset, although we couldn't compare to the exact same sort of muscle. This may point to common underpinning mechanisms driving, at least in fast-twitch fibers, skeletal muscle aging. Last but not the least, we employed aged muscle with no signs of sarcopenia per se, meaning that muscle mass was still intact. This is an extremely important point because it allowed us to probe for molecular changes that occur prior to the sarcopenia phenotype establishment, which takes this study to a less reactive and more preventive side of EDL muscle aging knowledge.

Together, we validated others findings but also provided novel ways that unraveled the negative side of mitochondrial alterations in fast muscles. We do believe that this will

be of value to the community, not only from the resource point of view, but also from the nature of aging mechanisms perspective.

1. As the title shown, the authors want to find the mechanism of fast myofiber loss during aging. Data shown EDL had mild tubular aggregation and may have smaller myofiber size, but no loss of muscle mass. Have any other evidence to support the EDL muscle loss during aging?

We thank the reviewer for raising this critical point. In fact, our evidence did not show any muscle loss (Fig. S1A) and here lies the critical point. The “old” time-point gave the leverage to investigate the molecular changes prior to the sarcopenia phenotype, it ensured that EDL was aged enough but hasn’t started to lose muscle mass. This is a narrow but instrumental window that enabled us to unfold mechanisms that may be driving later on muscle mass loss in fast-twitch fibers.

2. The old mice used in this study were 105 weeks (over 24 months old) should have severe sarcopenic phenotypes. Why authors mention that is non-sarcopenic mice?

We appreciate the reviewer’s concern about sarcopenic and non-sarcopenic states. According to the European consensus on the definition and diagnosis of sarcopenia (Cruz-Jentoft *et al*, Age Ageing, 2019), the newest definition of sarcopenia includes three different criteria: Low muscle strength; Low muscle quality and quantity; Low physical performance.

Clearly our mice, do not fulfill the criteria ‘low muscle quantity’, but we were able to observe qualitative (metabolic) changes. So clearly we explored some ‘pre-sarcopenic’ animals. In general, the reviewer is right, mice could be full blown sarcopenic at 24 months, but as already reported by others – they are not always. This depends clearly on the mouse strain, breeding strain or maintenance conditions. So, there are some reports in line with ours showing no changes in muscle weight up to 25 months old, and only at 28 months of age a loss of muscle was observed (Börsch A *et al*, Comm Biol., 2021).

Being aware of that, we carefully measured muscle mass and muscle cross sectional area (a marker of muscle quantity according to the consensus described) (Fig. S1A, Fig.1B and Fig.1C), which were here not affected by age. Furthermore, we show the quantifications of muscle mass normalized to tibia length as suggested by a different reviewer (Fig. S1A). Also these results were unaffected.

With the data obtained from this study we believe the mice used in the study were not yet sarcopenic, but already displaying early molecular changes that are likely to precede sarcopenia. This is crucial to understand what are the age-related changes prior sarcopenia to improve research outputs towards muscle health to delay or avoid the onset of sarcopenia.

3. This manuscript indicated that the dysfunction of mitochondrial respiration was due to the morphological defect of mitochondria. (1) The defects of mitochondrial morphology have bad effects on OXPHOS and glycolysis function, as well as most metabolic actions in mitochondria. (2) authors chose EDL to be the studied target. Most myofibers in EDL are type II. Why authors want to focus on the OXPHOS function? Have any special reason?

We thank the reviewer for highlighting this important aspect of our study. We understand that it may sound counterintuitive, but the reason why OXPHOS function was analyzed lies in the unbiased discover of mitochondrial pathways downregulation

(transcriptomics, Fig.2A) (also confirmed using sc-RNA-seq, Fig.S2B) and the enrichment for mitochondrial localization of EDL age-related metabolites (metabolomics, Fig.2C). After we found this unexpected overlap (using two different approaches) in a more glycolytic muscle, this array of results set us to investigate in more detail the mitochondria, in which we of course included OXPHOS (mitochondrial respiration). This strengthens the connection between omics data and the functional side of the observed age-related molecular changes. On the other hand we believe that the glycolytic type II muscle might be more susceptible to a loss of mitochondrial mass and function, since the initial mitochondrial amount is already low. Being glycolytic does not mean for EDL to be totally independent on mitochondrial function. It seems to be just more vulnerable to the loss of the (already low) function of mitochondria.

4. In this manuscript, authors compared the gene expression between mouse EDL and human gastrocnemius and found that they had similar expression patterns. However, these two parts of muscles were composed of different types of myofibers. If they had similar expression patterns at aged stage, does it have any meaning? Similarly, why different aged tissues/organs were positively associated with transcriptomic age-related changes? Different tissues/organs have different functions and regulation. If they have similar age-related changes, does it have any meaning?

Thanks for raising these interesting points.

- a) Regarding the similarity between mouse EDL and human gastrocnemius (GC): Although we cannot be entirely sure, we believe that the similarities found can be due to two main reasons. 1) the GC is composed by two small areas mostly oxidative and one bigger mostly glycolytic, being considered in the literature as a mostly glycolytic muscle (Mänttari S & Järvillehto M, BMC Physiol., 2005). The GTEx GC dataset results from the muscle RNA sequencing as a whole, and thus the similarity found between mouse EDL and human GC may be accounted for the glycolytic part of the human GC. 2) the other plausible explanation has to do with similar aging pathways observed across tissues, meaning that these mitochondrial processes downregulation is a commonly observed aging signature across tissues and different species. In fact, this is observed in our study (Fig.5b) and many other (Tyshkovskiy, A *et al.*, *Cell Metab.*, 2019; Tyshkovskiy A *et al.*, *Cell*, 2023; Amorim J.A *et al.*, *Nat. Rev. Endocrinol.*, 2022)
- b) Regarding the commonly observed aging signatures: We believe aging is a complex multifactorial process. The multi-omics revolution has shed light in recent years showing that different tissues lose their epigenome and transcriptome identity. And although some changes are tissue-specific, others are commonly observed not only across different tissues but as well as across different species. To name a few common changes, the upregulation in inflammatory signatures, disturbance in proteostasis and the downregulation of mitochondrial processes encompasses some of such conserved changes. The meaning of this commonality is still unclear, as the debate whether aging is caused by damage, failure in the continuation of developmental stages, or even stratified combination of both is still an active debate. Our study highlights mostly the importance of mitochondrial functional decline at pre-sarcopenic stages, but of course one cannot exclude other pathways activated in mid-age for example, that trigger the later observed mitochondrial dysfunction phenotype. We provide here an framework (pre-sarcopenic) that may use to engage mitochondrial protection before the critical muscle mass loss is occurring.

Reviewer #2 (Remarks to the Author):

Sarcopenia is mainly attributable to the atrophy of fast glycolytic fibers. Fernando et al. used gene expression and metabolite profiling of the extensor digitorum longus (EDL) muscles from 16- and 105-week old mice. Changes based on transcriptomic and metabolomic profile were further validated using various assays (oroboros mitochondria respiration, immunoblot, transmission electron microscopy). Commendably, additional public data sets were included in analyses. scRNAseq of Myh4 expressing nuclei was used to confirm that changes measured in intact EDL muscle are consistent with nuclei from type 2b fibers. Other publicly available datasets from human samples of various ages were compared with mouse datasets offering insights into the translatability of their findings with overlapping pathways. Interpretation was further extended by showing that life span extending interventions are negatively associated with changes occurring in EDL muscle. The results mostly confirm the findings of various individual studies across the literature but with the combination of metabolomic and transcriptomic analyses. The use of publicly available datasets to extend the findings was a strength of the study.

We thank the reviewer for such a positive feedback and encouragement of our work. Especially in seeing the value of the combination of metabolomic and transcriptomic analyses and the combination of the results with the public available datasets.

Minor comments:

- Changes in muscle mass would be better expressed normalized to TA length.

We agree with the reviewer's point. We added the following plot to the manuscript and change the results description and supplementary figure legend accordingly (highlighted in yellow).

Fig. S1 Aged EDL shows molecular damage but no loss of muscle mass. (A) The graphic represents the muscle weights standardized to tibia length of both young and old EDL muscle.

- Fast muscle fibers are not 'resistant to power and strength'

Thank you for this comment. We changed the phrase and the sentence is now more describing the metabolic properties of the muscle types:

The former is enriched in mitochondria, displaying an oxidative metabolism, while fast fibers rely mainly in glycolytic metabolism.

- MuRF1, Mafbx are better described as markers of atrophy than 'damage'. Damage frequently implies injury to cell membrane rather than breakdown of protein.

We thank the reviewer for pointing this out. Although, we do have a different view on the terminology of 'damage' (not only limiting it to membrane damage). We acknowledge that this might be confusing for some readers and replaces the word 'damage' to 'atrophy' throughout the manuscript.

At the gene expression level, there were increased levels of *Fbxo32* (MAFbx) and *Trim63* (MuRF1) (another marker of muscle atrophy),

Moreover, at the molecular level, we saw a strong increase in 3-methylhistidine metabolite in older muscle, as well as MAFbx (Atrogin-1) protein, a marker of protein breakdown and muscle atrophy, respectively¹¹ (Fig. 1D).

A possible explanation for CSA decline in aged EDL could be the increased muscle atrophy, considering the increased levels of muscle degradation (3-methylhistidine) and atrophy (MAFbx) markers

- Discussion states that this study set out to understand why fast glycolytic muscle fibers is susceptible to aging relative to slow oxidative muscle but the experiments did not compare fast and slow muscle.

Thank you for this remark. We corrected the statement to:

In the present study, we set out to understand why the fast glycolytic muscle is highly susceptible to age-related changes and what are the underlying metabolic mechanisms^{18,19}.

REVIEWERS' COMMENTS:

Reviewer #1 (Remarks to the Author):

Thanks for authors' responses. About these responses, please let me give you some comments as follows.

1. I agree the authors mentioned that "With the data obtained from this study we believe the mice used in the study were not yet sarcopenic, but already displaying early molecular changes that are likely to precede sarcopenia.", but this is not suitable for using non-sarcopenic mice to describe these aged mice.

2. As authors' responses, is it possible to hypothesize mouse and human had the similar molecular regulations during aging process. But it is not easy to understand how to compare the data between human and mouse, even authors gave us the supplemental figure 4. Different age comparisons were shown as human aging and mouse aging, but three comparisons were analyzed using different ages and groups. Or authors just wanted to show and analyze the final one?

3. I can understand why authors want to compare the data among different species and tissues/organs. In authors' responses, also mentioned about the reasons for this. However, I still suggest that authors can have some detail explanation about why these data can support the title indicated at the final paragraph of results.

Reviewer #2 (Remarks to the Author):

My comments were adequately addressed.